# S100 as Serum Tumor Marker in Advanced Uveal Melanoma

**DOI:** 10.3390/biom13030529

**Published:** 2023-03-14

**Authors:** Martin Salzmann, Alexander H. Enk, Jessica C. Hassel

**Affiliations:** Department of Dermatology and National Center for Tumor Diseases, University Hospital Heidelberg, 69120 Heidelberg, Germany

**Keywords:** uveal melanoma, S100 protein, serum tumor marker

## Abstract

**Simple Summary:**

Our study looked at the usefulness of a protein called S100 in monitoring the progression of uveal melanoma, a type of black skin cancer first growing in the eye. We found that only a small number of patients had high levels of S100 in their blood when the cancer had first spread to other organs. However, in over half of the patients, S100 levels increased as the disease progressed. Our study also looked at the expression of S100 in melanoma metastases, which are cancer cells that have spread to other parts of the body. We found that patients with strong expression of S100 in metastases always had rising levels of S100 in their blood during the course of the disease, while patients without S100 expression in metastases never showed rising S100 levels. Therefore, we concluded that S100 serum levels can be predicted by examining the expression of S100 in metastases and may be useful in monitoring disease progression, but it is not a reliable marker for early detection of the cancer.

**Abstract:**

S100 protein is routinely used as a serum tumor marker in advanced cutaneous melanoma. However, there is scarce and inconclusive evidence on its value in monitoring disease progression of uveal melanoma. In this monocenter study, we retrospectively assessed the connection between documented S100 protein levels of patients suffering from stage IV uveal melanoma and the clinical course of disease. Where available, we analyzed expression of S100 in melanoma metastases by immunohistochemistry. A total of 101 patients were included, 98 had available serum S100 levels, and in 83 cases, sufficient data were available to assess a potential link of S100 with the clinical course of the uveal melanoma. Only 12 of 58 (20.7%) patients had elevated serum levels at first diagnosis of stage IV disease. During progressive disease, 54% of patients showed rising serum S100 levels, while 46% of patients did not. Tumor material of 56 patients was stained for S100. Here, 26 (46.4%) showed expression, 19 (33.9%) weak expression, and 11 (19.6%) no expression of S100. Serum S100 levels rose invariably in all patients with strong expression throughout the course of disease, while patients without S100 expression in metastases never showed rising S100 levels. Thus, the value of S100 serum levels in monitoring disease progression can be predicted by immunohistochemistry of metastases. It is not a reliable marker for early detection of advanced disease.

## 1. Introduction

Uveal melanoma (UM) is a rare malignant tumor of the eye with an incidence of 5.1 cases per million per year [1], yet it is the most common intraocular malignancy in adults and accounts for about 5% of all melanomas [2]. About half of the patients are cured by local treatment [3], while the others will eventually metastasize, especially to the liver [4]. Even with a breakthrough in the treatment of cutaneous melanoma (CM) with checkpoint inhibitors [5] and targeted treatment [6], prognosis of advanced UM remains poor [7], with little response seen in conventional immunotherapies so far [8]. Recently, tebentafusp was approved for treatment, a bispecific protein that prolongs patient survival significantly [9]. Even patients with radiologically progressing tumors may benefit.

S100 is a family of calcium-binding proteins that are primarily found in nerve cells and glial cells. However, some S100 proteins have been found to be elevated in the blood of patients with certain types of cancer and may serve as serum tumor markers [10]. For example, S100B is a protein that is normally found in the brain, but it can also be produced by certain types of cancer, such as melanoma and glioma [11]. Other cancers, such as breast cancer [12], colorectal cancer [13], and lung cancer [14], may be associated with elevated levels of other members of the S100 protein family, such as S100A4. Also, their role in inflammatory diseases is increasingly reported [15,16]. The use of S100B, in this manuscript abbreviated as S100, is common as a serum tumor marker for cutaneous melanoma and has been established for a long time [17]. Its expression on melanoma cells dates back to a first publication in 1980 [18]. S100 measurements are now universally recommended in national and international guidelines on cutaneous melanoma [19] and belong to commonly available and routinely performed laboratory tests in the follow-up of melanoma patients. While the use of S100 in cutaneous melanoma is well established, there is scarce and inconclusive evidence on the performance of S100 protein as a serum tumor marker in advanced uveal melanoma: while some studies suggest a link with tumor burden [20,21,22,23], Strobel et al. question its value and sensitivity [24]. Of note, none of these studies included more than 32 patients in a metastasized stage. With S100 measurements routinely performed at many institutions, there is currently a lack of evidence on its performance in predicting progression of UM patients. Even though novel biomarkers may be more promising in the future [25,26,27,28,29], the value of S100 as a serum tumor marker in UM is still a relevant issue of daily patient care due to its widespread availability.

Similarly to the available literature on serum S100, few studies have reported on the expression of S100 in uveal melanoma. Available studies suggest an expression of S100 in about 66% of metastases [30], with a strong expression in less than 50% [31].

The aims of our analysis were to assess the frequency of S100 protein elevation in patients with advanced UM, to find a connection with the clinical course of the disease, and to correlate serum S100 elevation with S100 expression of metastases.

## 2. Materials and Methods

### 2.1. Patients

This was a monocenter, retrospective study performed at the Department of Dermatology and National Center for Tumor Diseases at the Heidelberg University Hospital. We systematically included all patients treated for stage IV UM at our institution with digital records available from 2010 to November 2022. Treatment was defined as any systemic anticancer treatment, liver-directed local treatment, or supportive treatment. Patients were excluded if they were seen only for a second opinion and treated elsewhere. There were no further exclusions made from the cohort. Missing values are reported in the results section wherever applicable.

### 2.2. Data Collection

All patients were treated in routine clinical practice and not for the purpose of this study. Clinical records were screened retrospectively and data merged centrally. All available values of S100 and LDH were documented. Treatment lines and dates of progression, as well as data on survival were documented. We also reviewed available histopathological staining of metastases for their expression of S100 protein, gp100 (HMB-45) [32] and melan A [33]. Where available, expression of S100 was evaluated by H-score [34]: A strong expression was defined as H-scores of 2 and 3, a weak expression was defined as an H-score of 1, and no expression was defined as an H-score of 0. External histopathological reports were included only if the description on S100 staining gave clear evidence on the respective grading by H-scores (e.g., strong expression, weak expression).

### 2.3. Statistical Analyses

Statistical analysis was performed using IBM SPSS Statistics, version 27.

Time points of available imaging (CT and MRT scans as clinically indicated, routinely performed every three months), available S100 serum levels and available LDH levels were collected. Radiographic response to treatment, including the criteria for progressive disease, was defined by RECIST criteria version 1.1 [35]. A connection of S100 levels with tumor progression was assumed with rising levels concurrently with rising LDH serum levels and radiologically confirmed progression. A connection of S100 levels with the disease was ruled out if radiographic progression and pathological elevation of LDH levels were observed with no increase in S100 levels.

Overall survival (OS) was calculated from first diagnosis of stage IV disease until death from any cause. In patients alive at time of final data analysis as well as in patients lost to follow-up, the date of last contact was used for censored calculation. Survival was estimated by the Kaplan–Meier method. Univariate comparisons of Kaplan–Meier estimators were performed using the log-rank test. Two-sided Fisher’s exact and chi-squared tests were used for comparisons among groups with categorical variables. Multivariate analyses were performed by Cox regression. *p* values were considered significant at *p* < 0.05.

### 2.4. Ethical Approval

Retrospective analyses of clinical data were approved by the institutional review board of the Medical Faculty of the University Hospital Heidelberg (S-454/2015). The ethics committee had agreed to the retrospective analysis of routinely collected clinical data without prior informed consent of patients.

## 3. Results

### 3.1. Patient Characteristics

A total of 101 patients were included, 56 female (55.4%) and 45 male (44.6%). Median age at diagnosis of stage IV disease was 63 years (range 17–89). Patients had a median OS of 17.3 months (range 1.4–100.5 months), and 27 were alive at the time of final follow-up in November 2022. Most patients (96, 95%) suffered from hepatic metastases. Patients received a median of two systemic lines of treatment (range 0–6), but only 10 patients (9.9%) responded to any kind of systemic treatment. 10 patients (9.9%) were in an ongoing remission after excision, local or systemic treatment of metastases. A summary of patient characteristics is shown in Table 1.

### 3.2. Serum S100 Elevation

Over the course of the disease, a median of three S100 serum levels were assessed per patient (range 0–19). Any S100 level measurement was available in 98 patients (97%). Based on treatment duration and survival, the number of assessed values differed between patients: 25 (24.8%) had a single available value, 42 (41.6%) had two to four available values, 21 (20.8%) had five to seven available values, and 10 (9.9%) had eight or more available values (one patient had 8, three patients had 9, one patient had 10, one patient had 11, two patients had 13, one patient had 15, and one patient had 19 available values). Any S100 elevation was observed in 53 patients (54.1%); in median, the highest measured S100 serum level was 2.78 xULN (upper limit of normal) (range 1.04–219.86) in these patients. In sum, 58 patients (57.4%) had available S100 levels at the first diagnosis of stage IV disease. Of these, 12 (20.7%) had an elevated S100 level and 46 (79.3%) had normal values at first diagnosis of stage IV. Of the patients with an S100 elevation throughout the course of disease, 12 of 31 (38.7%, 23 missing values at first diagnosis) had an elevation already, while 19 of 31 (61.3%) had normal values at first diagnosis of stage IV.

### 3.3. Connection of Serum S100 with Tumor Progression

To assess the connection of S100 levels with the clinical course of the uveal melanoma progression, time points of radiologically confirmed progression with available S100 and LDH serum levels were collected. Increasing serum S100 (or pathologically elevated S100 levels in patients with only one value available) during progressive disease and concurrent increase of LDH levels were detected in 45 of 83 patients (54.2%) with available data at time of progression. In 38 of 83 patients (45.8%), no increase in S100 levels was observed during radiographically confirmed progression and rising pathological LDH levels, and thus there was no connection between S100 serum levels with the clinical course of the uveal melanoma. In 18 patients, there was no relative time point available for evaluation. This was due to lack of baseline values or lack of disease progression.

There was no statistically significant difference between the groups of patients with or without S100 connection to disease status regarding age (*p* = 0.590), time between treatment of the primary to stage IV disease (*p* = 0.096) and OS since first diagnosis of stage IV (*p* = 0.194).

A summary on S100 serum levels and the connection of serum S100 with tumor progression is given in Table 2.

### 3.4. Expression of S100 in Uveal Melanoma Metastasis and Correlation of Expression with Serum S100 Levels

A total of 57 in-house histopathological stains and 16 external histopathological reports of metastases were available (73/101, 72.3% of patients), of which 44 (60.3%) samples were collected to verify the first diagnosis of stage IV disease and 29 (39.7%) were taken at a later stage. Overall, 41 metastases were liver biopsies (56.2%), 23 were subcutaneous metastases (31.5%), three were bone metastases (4.1%), while other localizations (6 samples, 5.9%) included lung metastases (two), metastasis of the small intestine, malignant pleural effusion, brain metastasis, and peritoneal metastasis.

Of 56 available S100 stains, 26 showed a strong expression (as defined by H-score of 2–3) and 19 a weak expression (46.4% and 33.9%, respectively) (Figure 1). No expression was observed in 11 samples (19.6%). Melan A was expressed in 50 of 51 tumors (98%), Hmb45 in 57 of 60 tumors (95%).

The elevation of S100 in patient serum correlated significantly with the expression of S100 in the tumor, as shown in Figure 2. In median, the highest measured serum S100 per patient was 0.60 xULN in patients with no expression of S100 in metastases (range 0.40–0.78 xULN), 0.91 in patients with weak S100 expression of metastases (range 0.28–5.31 xULN), and 2.77 xULN in patients with strong expression of metastases (range 0.51–46.96 xULN).

In sum, 44 patients had both available tumor tissue and sufficient data to assess the connection between the course of serum S100 levels and disease progression. Table 3 demonstrates whether S100 levels increased with disease progression based on S100 expression: If S100 were expressed by metastases, serum S100 levels rose in all 21 patients during the course of progressive disease. On the other hand, if S100 were not expressed by metastases, S100 serum levels stayed normal even during LDH elevation and radiologically confirmed progression. In patients with weak expression of S100, only 3 of 14 patients developed increasing S100 serum levels throughout the course of the disease (*p* < 0.001).

## 4. Discussion

S100 protein has been used in UM as a serum tumor marker due to its similarity to CM by many physicians, with currently inconclusive studies on its actual value to predict progression of UM. In clinical routine, it may be used in the setting of early detection in the absence of metastases or to monitor the course of disease in patients already suffering from advanced disease. False-negative or false-positive results may lead to wrong treatment decisions. Thus, the aim of our study was to determine whether serum S100 is associated with the course of disease in UM patients, and to potentially link these serum tumor levels to the expression of S100 in the tumor.

In our cohort, only 54% of advanced UM patients ever showed elevated S100 serum levels; conversely, almost half of the patients never showed an increase. The same ratio was found in our analysis to assess the connection between rising S100 serum levels and progression of UM, as defined by rising LDH levels and radiographically confirmed progression. Overall, our data show that S100 protein was a functional tumor marker only for about half of our patients with advanced UM.

Even in the patients with rising S100 levels throughout the course of disease, only 39% already showed increased S100 values at the time of first clinical detection of metastases. Among our entire cohort, only 21% had elevated serum S100 at the first diagnosis of metastases. This shows a major limitation of using serum S100 in tumor aftercare after primary resection, as S100 levels may not be elevated at all or only late in the course of disease. In addition, false-positive results are common [36]. S100 serum levels are thus not suitable for reliable early detection of metastases, an utterly important function of any tumor marker [37]. In this situation, with no metastases present, a confounder may lie in a difference in S100 expression between the primary stage and metastasis, which has been described previously [30].

However, we show that its value in monitoring advanced disease can be predicted by immunohistochemistry of metastases. Whenever there was a strong expression of S100 protein by metastases, its serum levels rose invariably during the course of disease. On the other hand, none of the patients without expression of S100 showed an increase during the course of disease, and thus there is no rationale in measuring these serum levels. For patients with weak expression of S100, an increase was observed in a minority of patients, potentially indicating a high necessary tumor load to lead to an increase in serum levels. Thus, though S100 protein is a functional serum tumor marker for UM expressing S100, it is unusable as a serum tumor marker for UM not expressing S100, and has limited value in UM with weak expression of S100. As a limitation to this finding, the use of this correlation is only possible in patients with already diagnosed stage IV disease, and not during follow-up after primary treatment only.

In daily practice, our data may help the clinician to better put available values into perspective. Especially a false sense of security should be avoided with normal S100 levels, as there may never be an increase during the course of the disease. On the other hand, S100 serum levels may be valuable information in monitoring disease progression in tumors with an expression of S100.

We are aware of many limitations of our study, which mainly include its retrospective nature, limited patient numbers, and nonstandardized monitoring of data during clinical routine.

## 5. Conclusions

Serum S100 protein levels are linked to disease progression in about half of advanced UM patients; however, S100 performed poorly in early detection of metastases in our collective. Its value is closely linked to expression of S100 protein by metastases, and thus it can be predicted by immunohistochemistry.

## Figures and Tables

**Figure 1 biomolecules-13-00529-f001:**
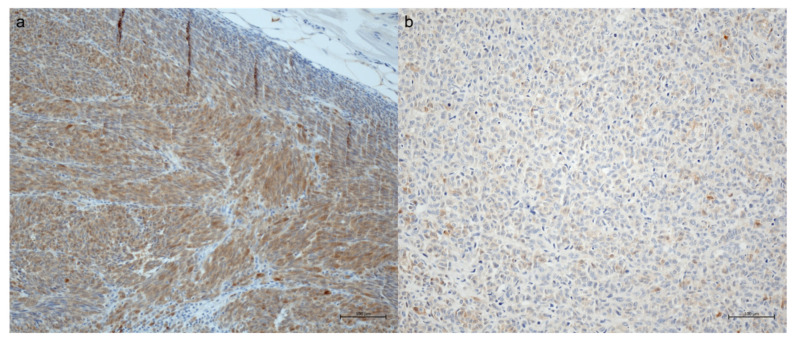
S100 stains of metastases, (**a**) with strong expression (H-score 3), scale bar: 100 µm; (**b**) weak expression (H-score 1), scale bar: 100 µm.

**Figure 2 biomolecules-13-00529-f002:**
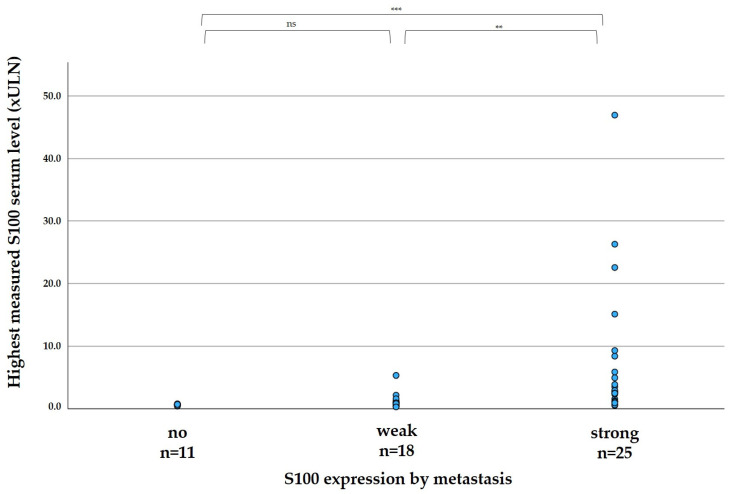
Highest measured S100 serum levels compared between the subgroups of patients with metastases showing no expression, weak expression or strong expression of S100. 56 patients had S100 stains available, two patients were excluded with no S100 serum levels obtained. ns: not significant, *p* = 0.402 ** *p* = 0.001, *** *p* < 0.001.

**Table 1 biomolecules-13-00529-t001:** Patient characteristics. * Other liver directed treatment included chemoperfusion in 6 patients, selective internal radiation therapy (SIRT) in 4 patients and transarterial chemoembolization (TACE) in 2 patients.

Parameter	Number of Patients	
		(%)
Total number of patients	101	100
Age (years), median	63 (range 17–89)	
Gender		
Male	45	44.6
Female	56	55.4
Overall survival (months), median	17.3 (range 1.4–100.5)	
Disease status		
Patients deceased	74	73.3
Patients alive	27	26.7
Tumor in ongoing remission	10	9.9
Localization of metastases		
Hepatic	96	95.0
Pulmonal	50	50.5
Osseous	38	37.6
Subcutaneous	31	30.7
Other	48	47.5
Number of systemic treatment lines		
0	9	8.9
1	29	28.7
2	35	34.7
>2 (max. 6)	28	27.7
Response to systemic treatment	10/92	10.9
Checkpoint inhibitors	3/71	4.2
Chemotherapy	1/32	3.1
Tebentafusp	6/29	20.7
Liver-directed treatment (LDT)	40	39.6
Chemosaturation	28/40	70.0
Other liver-directed treatment *	12/40	30.0
Response to LDT	3/40	7.5
Stable disease after LDT	15/40	37.5

**Table 2 biomolecules-13-00529-t002:** Characteristics of S100 serum levels.

Parameter	Number of Patients	
		(%)
Total number of patients	101	100
S100 serum levels available (median)	3 (0–19)	
0	3	3.0
1	25	24.8
2–4	42	41.6
5–7	21	20.7
8–19	10	9.9
Any serum S100 levels available	98	97.0
elevated at any time	53/98	54.1
never elevated	45/98	45.9
S100 serum levels available at first diagnosis of stage IV	58	57.4
elevated	12/58	20.7
normal	46/58	79.3
Sufficient data available for assessment of connection between S100 and clinical progression	83	82.2
connection	45/83	54.2
no connection	38/83	45.8

**Table 3 biomolecules-13-00529-t003:** Dependence of serum S100 increase on tumor S100 expression. The difference between the groups was statistically significant (*p* < 0.001).

		Serum S100 Increase during Disease Progression			
		No		Yes	
		*n*	%	*n*	%
Tumor S100 expression	None (*n* = 9)	9	100	0	0
	Weak (*n* = 14)	11	79	3	21
	Strong (*n* = 21)	0	0	21	100

## Data Availability

The datasets used and/or analyzed during the current study are available from the corresponding author on reasonable request.

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
