# Peer review of "S100 as Serum Tumor Marker in Advanced Uveal Melanoma"

_biomolecules, 2023, doi:10.3390/biom13030529_

Round 1

Author Response

Dear Reviewer,

please find our revision letter in the attachment. Thank you!

Reviewer 2 Report

S100 as a serum tumor marker in advanced uveal melanoma

The paper submitted for review is an attempt to determine whether S100 can be applied as a marker for advanced uveal melanoma (UM). The authors compared serum S100 levels with the findings of immunohistochemical testing in metastatic tumor tissues in cases of stage IV UM. The experimental idea was perhaps a good one, and the authors took a reasonable approach to it. The findings of the report suggest, however, that S100 is not a particularly valuable marker for advanced UM, especially in the area of early detection of metastases. 

The introduction of the report introduces UM as a rare malignant tumor of the eye, though in fact it is the most common intraocular malignancy in adults and also accounts for 3 - 5% of all melanomas. It is, however, much less frequent than many other neoplastic diseases and scarcity of archived histopathological material makes assessments of this type all-the-more difficult. The authors themselves state that there are few reports available on the expression of S100 in UM and their present study is, by their own admission, of rather low strength owing to limited patient numbers. 

The materials collected included data regarding, amongst other things, serum S100 and LDH levels, as well as the results of immunohistochemistry tests in histopathological diagnoses. The authors also collected data regarding patient age, treatment methodologies, overall survival time, and response to treatment. The statistical approach was of standard form and did not raise any concerns.

The results section began by telling us that the cases of 101 patients were included in the study. It was not entirely clear though whether there were some exclusions made from the cohort. Table 1 reports on 101 patients in total but Table 2 shows that 3 of these had no serum S100 results available and section 3.4 of the report said that while there were 57 sets of histopathological stains available for the correlation tests, only 56 of these included an S100 stain. It is also not clear how the 16 external histopathology results were used in the correlation testing.

S100 is not one protein but rather, a large family of of proteins with more than 20 known members. The protein has been in use in the diagnosis of melanomas for a very long period as some of the cited research clearly demonstrates (the authors have cited work from 20, and even from 30 years ago). The S100 name was first used in the 1960s and since that time, research has demonstrated that the many different proteins in this group are actually involved in a wide range of cancers in vertebrates, not just in melanomas. It would be worthwhile making that clear in this report. It is known, for example, that S100A4 is frequently positive in lung, prostate, and colorectal cancers, as well as in melanoma.

The S100 group has been extensively tested over the years. This causes me to wonder why the authors of the study chose this particular approach at this point in time. In recent years, developments in the area of melanoma research have introduced other markers such as BAP1 expression, chromosome 3 monosomy, and the increased or decreased expression of certain miRNAs. Some explanatory comments in the paper would be of value, to show why the authors believe this research remains relevant.

The authors wrote that clinicians should avoid a false sense of security on receiving a normal S100 serum result, which is a very valuable statement. On the other hand, they also wrote that S100 serum tests are unusable in cases of UM where S100 is not expressed. How then, is a clinician to decide if S100 testing is relevant? Since the results of serum testing were highly correlated with immunohistochemistry tests, the authors say that they have shown the value of monitoring advanced disease "can be predicted by immunohistochemistry of metastases." I would respectfully suggest, however, that the reverse might be true - that is, positive immunohistochemistry might be predicted by serum results. It is unfortunate that the authors suggest making predictions on the basis of immunohistochemistry in metastatic tumor tissue, as the fact of its existence means the patient is already unlikely to survive. UM patients treated for primary local disease stand a good chance of survival (around 60-70% will survive 5 years) while patients suffering from metastatic disease have around an 8% chance of surviving 2 years.

The reference material is, as previously stated, rather dated in parts. Some approaches to diagnosis and treatment have changed over the course of the last 20 - 30 years. I would like to see the references bolstered and updated. While the information presented remains true other approaches, some of which I have alluded to above, have been introduced more recently and show promise. On the other hand, some of the reference material, including the use of immunotherapy as a treatment strategy is very much up-to-date.

Use of English is good and only a minor review will be necessary - preferably by a native speaker who has not previously seen the paper as they are more likely to spot the few small errors.

Overall, the authors took a reasonable experimental approach to a "vintage" marker, though the findings were not as promising as they may have hoped I applaud their determination to publish what some would see as a negative result. Prior to publishing though, the work should be reviewed to include some more up-to-date background on the S100 family and on the diagnosis of melanomas, including some of the current alternative strategies. The authors should also clarify how much of the collected data was excluded as incomplete or incomparable.

Author Response

(The authors gave the same response as above.)

Reviewer 3 Report

Dear authors,

It was with great interest I read the manuscript exploring the possibility to use S100 as a marker for early disease detection and to monitor progression in patients with uveal melanoma. The manuscript contains important information, especially that S100 only works in patients with tumors expressing S100. I have some comments to clarify the interpretation of the presented data, as of now it is a little hard to follow all the results and conclusions.

1.     Introduction, Line 29 concerning tebentafusp, remove “with” and rewrite the sentence.

2.     Clarify the aims, as I see it is three aims (that you can potentially change order in) change order in). “The aim of our analysis was (1) to correlate serum S100 elevation with S100 expression in the metastastis, (2) to assess the frequency of S100 protein elevation in patients diagnosed with metastatic UM, (3) to correlate S100 protein elevation with tumor progression.

3.     Methods: How progression was defined is unclear to me, was it necessary to both have radiological signs of progression AND increased levels of LDH (that is how I get the current writing at least)? That seems as a strict definition, not all patients have increasing LDH levels, but still have obvious progression on radiology?

4.     Results: Only systemic therapies are reported, but it seems like patients also have undergone surgery or loco-regional treatments (you write excision, local and systemic). Report also these treatments in Table 1.

5.     I recommend a rearrangement of the results to ease reading. E.g., start with the first paragraph and Fig 2 in the section “3.5 Correlation of tumor S100 expression and serum S100 levels”, then continue with the first part of paragraph in 3.3. Then a new paragraph with all data for aim 2, and then the same for aim 3. As of now, the aims are dispersed through-out the Results making it hard to follow.

Author Response

(The authors gave the same response as above.)

Round 2

Reviewer 2 Report

After answering all of my questions and updating the text of the article appropriately, I feel that the authors have satisfied all of my doubts and misgivings. The paper will still require a linguistic review, particularly where text has been added or changed. Aside from that, I am now happy that the article is suitable for publication.